# The Mitochondrial Genomes of *Siboglinum plumosum* and *Oligobrachia dogieli* (Annelida: Siboglinidae) and Their Phylogenetic Analysis

**DOI:** 10.3390/genes15010077

**Published:** 2024-01-07

**Authors:** Elizaveta K. Skalon, Zinaida I. Starunova, Sergey A. Petrov, Roman V. Smirnov, Olga V. Zaitseva, Viktor V. Starunov

**Affiliations:** Zoological Institute, Russian Academy of Sciences, 199034 Saint-Petersburg, Russia; elizaveta.skalon@zin.ru (E.K.S.); zinaida.starunova@zin.ru (Z.I.S.); sergey.petrov@zin.ru (S.A.P.); roman.smirnov@zin.ru (R.V.S.); olga.zaitseva@zin.ru (O.V.Z.)

**Keywords:** Siboglinidae, frenulates, mitochondrial genome, Sea of Okhotsk

## Abstract

Frenulates are a group of sedentary Annelida within the family Siboglinidae that inhabit the ocean floor and present a unique challenge for comprehensive molecular and phylogenetic investigations. In this study, we focused on the frenulates, specifically assembling the mitochondrial genomes of *Siboglinum plumosum* and *Oligobrachia dogieli*. The phylogenetic reconstruction placed *S. plumosum* as a sister taxon to *S. ekmani*, and *O. dogieli* as a sister taxon to *S. fiordicum*, supporting the non-monophyletic nature of the genus *Siboglinum*. Overall, this study supports the phylogeny of the family Siboglinidae while highlighting the need for additional molecular data within frenulates.

## 1. Introduction

Siboglinids, commonly known as beard worms, are a sedentary group of marine annelids with highly specialized adaptations enabling them to inhabit the ocean floor. The taxonomic and phylogenetic position of this group has a rich history spanning over a century. The most recent investigations have confirmed Siboglinidae as a family of Annelida within the Sedentaria subclass [1,2]. The family Siboglinidae comprises more than 200 species in 33 genera (World Register of Marine Species at https://www.marinespecies.org/aphia.php?p=taxdetails&id=129096, accessed on 23 December 2023). “Pogonophora” is a term historically used to describe a group, now encompassed within the broader clade Siboglinidae. “Pogonophora” and “Frenulata” have sometimes been used interchangeably, but both groups are currently not accepted in the WoRMS database. Nevertheless, “frenulata” is still employed in the literature as a clade name within the Siboglinidae family [3,4]. The position of different rank-free groups within Siboglinidae has undergone significant changes over time, highlighting the dynamic nature of siboglinid taxonomy.

Siboglinids possess unusual morphology that distinguishes them from other annelid families. These tube-dwelling worms lack a mouth and anus, and instead, rely on the absorption of nutrients directly through the body wall. They have long tentacles (or a single one) designed for nutrient absorption from the surrounding water and a specialized feeding structure known as a “trophosome” [4]. This special parenchymal organ is filled with symbiotic bacteria providing essential nutrients to the worm. The reliance on chemoautotrophic sulfur-oxidizing bacteria as a primary energy source allows siboglinids to inhabit extreme deep-sea muddy environments, such as hydrothermal vents and hydrocarbon cold seeps [3,4]. The associations between frenulates and their symbionts can vary based on the host species and on the habitat and geographic location of the host [4]. Conducting any research on these enigmatic animals is a challenging task due to the exceptional difficulty in collecting specimens and the consequent scarcity of material for both morphological and molecular studies. Furthermore, research vessels usually lack specialists with sufficient expertise to perform the required manipulations for proper fixation [5]. As a result, more than 70 years after their discovery and description, our knowledge about siboglinids still has many gaps.

The genus *Siboglinum* has 72 species representing over half of the diversity in the frenulates. Despite morphological diversity, a shared characteristic across all *Siboglinum* species is the presence of a single tentacle, with limited variations. *Siboglinum plumosum* Ivanov, 1957, like most members of the genus *Siboglinum*, typically possesses a single tentacle. Many species of siboglinids have been known for a long time and have detailed morphological descriptions. At the same time, molecular systematics has very scant data for phylogeny. Molecular analyses, including the examination of 16S mitochondrial ribosomal RNA (rRNA) and 18S nuclear rRNA genes, have been conducted only on two siboglinid species, *Siboglinum ekmani* and *Siboglinum fiordicum*. The non-monophyletic relationship observed in the molecular analyses suggests that *S. ekmani* and *S. fiordicum* are not closely related in an evolutionary context, challenging their grouping within the same taxonomic category [6]. 

The genus *Oligobrachia* comprises 11 species, ranking as the second largest after the genus *Siboglinum* among frenulates. Detailed information regarding the biology of *Oligobrachia dogieli* Ivanov, 1957 may be somewhat limited, but the morphological description includes a number of important details. *O. dogieli* most often possess more than a single tentacle (usually 3–9 tentacles) [7]. Notably, there is an absence of specific molecular data pertaining to *O. dogieli* within the public domain. Though morphological characteristics play a significant role in taxonomic classification, confirmation through molecular taxonomy is recognized as essential for a comprehensive understanding of the evolutionary relationships within this group.

While there is a large amount of morphological evidence on the structure of siboglinids, the molecular data, especially on frenulates, are very limited and still available for just a few species, making them inadequate for inferring phylogenetic relationships within this group (see e.g., [1,8,9,10]). Consequently, a comparative morphological approach and cladistic analysis relying almost exclusively on morphological data continues to be employed in constructing the frenulate taxonomic system. According to Ivanov [11], Webb [12], Southward [13], and Rouse [14] the genera *Siboglinum* and *Oligobrachia* occupy positions on the frenulate phylogenetic tree within two different clades of a family value (Siboglinidae s. str. and Oligobrachiidae, respectively) within a clade previously designated as the order Athecanephria. The taxonomic structure of the genus *Siboglinum* is still very complex and obscure. The only attempt at a detailed revision of the genus based on a morphological approach was presented by Smirnov [5]. He proposed the subgeneric classification of the genus with nine subgenera, including three monotypical ones, and determined the taxonomic significance of their morphological characters. 

Mitochondrial genomes are frequently used for phylogenetic analyses in metazoans due to their conserved nature and the information they provide about evolutionary relationships. To date, complete mitochondrial genomes have been publicly available for only four frenulate species [15]. Only two species from the genus *Siboglinum* are among them: *S. ekmani* and *S. fiordicum*. Both morphological and molecular approaches can provide insights into the unresolved issues and ongoing discussions in the field. Here, we report the mitochondrial genomes of two siboglinids, *S. plumosum* and *O. dogieli*, using phylogenetic analysis. We present a molecular phylogenetic assessment of two frenulate species which may give new insights into the main evolutionary trends in this peculiar group.

## 2. Materials and Methods

### 2.1. Sample Collection and DNA Extraction and Sequencing

Specimens of *S. plumosum* and *O. dogieli* (Figure 1) were collected in August 2022 in the Sea of Okhotsk off the eastern coast of Sakhalin, 51°32′05.3″ N 144°25′03.7″ E. The samples were taken using a Van Veen bottom grab from a depth of 270 m. The specimens were fixed in 96% ethanol and transported to the laboratory. The fixed samples were identified by Roman V. Smirnov. The photographs were taken using an Olympus MVX 10 macroscope (Olympus, Tokyo, Japan). All sampled material is deposited at the Zoological Institute of the Russian Academy of Sciences, St. Petersburg under the catalog numbers ZIN No. HN1 (*S. plumosum*) and SN37 (*O. dogieli*).

The total DNA was extracted using an ExtractDNA Blood & Cells kit (Evrogen, Moscow, Russia, BC111M) and stored at −20 °C. NGS libraries were prepared using the NEBNext Ultra II DNA Library Prep Kit for Illumina and NEBNext Multiplex Oligos for Illumina Dual Index Primers Set 1 (New England Biolabs, Beijing, China). The purification and concentration of resulting indexed libraries were performed using AMPure XP beads (Beckman Coulter, Beverly, MA, USA). The quality control of the resulting NGS libraries was performed using the QIAxcel Advanced System (Qiagen, Hilden, Germany).

Raw reads were generated using an Illumina NovaSeq 6000 instrument (Illumina, San Diego, CA, USA) at the Evrogen Core Sequencing Centre (Evrogen Company, Moscow, Russia). Before sequencing, the pooled libraries were once again controlled using the Agilent 2200 TapeStation System (Agilent, Boulder, CO, USA). The lengths of the reads were 2 × 150 bp. 

### 2.2. Assembly and Annotation

The quality of the paired-end read data was manually assessed using FastQC v0.11.5 (http://www.bioinformatics.babraham.ac.uk/projects/fastqc/, accessed on 16 November 2023). Sequencing adaptors, low-quality nucleotides, and reads with lengths less than 25 nucleotides were removed using Trimmomatic v0.36 [16] (ILLUMINACLIP:$ADAPTERS:2:30:10:2:TRUE SLIDINGWINDOW:4:20 MAX- INFO:50:0.8 MINLEN:25). The clean reads were de novo assembled using a GetOrganelle v1.7.1 pipeline [17]. The resulting mitogenomes were annotated with MITOS [18] using the Invertebrate Mitochondrial Genetic Code (NCBI, 5). Gene boundaries were manually refined using MAFFT v7.52 [19] and AliView v1.28 [20] by alignment against 15 published mitogenome sequences of Siboglinidae downloaded from the NCBI Nucleotide database (https://www.ncbi.nlm.nih.gov/, accessed on 16 November 2023): *Escarpia spicata* (ON929994.1), *Oasisia alvinae* (KJ789164.1), *Riftia pachyptila* (ON929992.1), *Alaysia spiralis* (ON929998.1), *Arcovestia ivanovi* (ON930000.1), *Ridgeia piscesae* (NC_024653.1), *Tevnia jerichonana* (NC_026862.1), *Seepiophila jonesi* (NC_026861.1), *Lamellibrachia columna* (ON929995.1), *Sclerolinum brattstromi* (NC_026855.1), *Osedax rubiplumus* (MT108937.1), *Siboglinum fiordicum* (KJ789170.1), *Siboglinum ekmani* (KJ789169.1), *Spirobrachia* sp. YL-2014 (KJ789171.1), and *Galathealinum brachiosum* (NC_026857.1). Mitogenome maps were generated using CGView (https://proksee.ca, accessed on 23 December 2023) [21].

### 2.3. Phylogenetic Analysis

The phylogenetic relationships of two newly obtained species, *S. plumosum* and *O. dogieli*, and 15 other siboglinids were reconstructed using 13 protein-encoding genes (PEGs) (see NCBI accession numbers in Section 2.2). Three annelid species, *Owenia fusiformis* (NC_028712.1), *Lumbricus terrestris* (NC_001673.1), and *Sabella spallanzanii* (NC_056279.1), were chosen as an outgroup. PEGs were either obtained from new assemblies or retrieved from GenBank using the python Bio.Entrez v1.81 package and were aligned using MAFFT v7.52 (L-INS-i method). Individual gene alignments were concatenated using SeqKit v2.4 [22], resulting in a supermatrix containing 11,479 amino acids. IQ-TREE v2.2.2.7 [23] with 10,000 ultrafast bootstrap replicates [24] was used to conduct a maximum likelihood (ML) analysis of the dataset partitioned by the gene. Nucleotide substitution models for each partition were determined by ModelFinder [25] under the Bayesian information criterion (BIC) implemented in IQTREE (Table 1). We also performed ML analyses with two additional datasets, one with the third codon position of each gene partitioned separately from the first and second positions, and the other with each of the codon positions partitioned separately. These analyses had the same topology as the original analysis; hence, for the discussion and further Bayesian inference (BI) analysis, we used only the dataset partitioned by gene. BI analysis was carried out using MrBayes v3.2.7a [26] on the CIPRES Science Gateway web server v3.3 [27] with two independent runs of four Markov Chain Monte Carlo (MCMC) chains running for 10 million generations and sampling every 1000 generations. To each partition, we assigned the closest evolutionary model available in MrBayes (Table 1) from those identified previously using ModelFinder. The MCMC trace file was analyzed with Tracer v1.7 [28]. Two runs converged, and all ESS values exceeded 7000. Burn-in was chosen to be 10% according to trace plots. Separate single-gene phylogenetic analyses were performed using the newly sequenced COX1 and 16S data for *S. plumosum* and *O. dogieli* and data for other members of *Lamellisabella*, *Spirobrachia*, *Siboglinum*, *Galathealinum*, *Polybrachia*, *Oligobrachia*, and *Bobmarleya* available from GenBank (Appendix A). *O. rubiplumus* was chosen as an outgroup. Sequences were aligned using MAFFT (L-INS-i method), and the resulting alignments were used to construct the ML trees in IQ-TREE with 10,000 ultrafast bootstrap replicates. The trees were visualized in FigTree v1.4.2 (http://tree.bio.ed.ac.uk/software/figtree/ accessed on 23 December 2023).

## 3. Results

The complete mitogenome of *S. plumosum* is a closed circular molecule of 15,291 bp in length (NCBI nucleotide accession number OR551480.1) (Figure 2). The average coverage of the assembled mitochondrial genome was 709.6x. The overall base composition of the mitogenome was 30.13% A, 10.80% G, 20.37% C, and 38.70% T, with a GC content of 31.17%. The mitogenome contains 37 genes including 13 protein-encoding genes (PEGs), 22 transfer RNA genes (tRNAs), and 2 ribosomal RNA genes. The putative secondary structures of ribosomal and transfer RNAs are presented in Appendix A. All the genes are transcribed along the forward direction. The length of tRNA genes ranges from 52 bp for trnR^tcg^ to 74 bp for trnE^ttc^. Some 14 genes are overlapped with their neighbors. The length of gene overlaps is from 1 to 8 bp, respectively (29 bp in total). The intergenic regions range between 0 and 597 bp (1161 bp in total). The longest intergenic region is located between trnR^cga^ and trnH^cac^ and may correspond to the control region.

For *O. dogieli*, an uncompleted mitochondrial genome was assembled (OR804078.1) (Figure 2). The average coverage of the assembled mitochondrial genome was 997.5x. All 37 genes (13 PEGs, 22 tRNAs, and 2 rRNAs) were complete. The uncompleted part corresponds to the control region (between trnR^cga^ and trnH^cac^). The total length of the mitogenome was 16,236 bp. The base composition was 29.74% A, 18.03% G, 9.95% C, and 42.28% T, with a GC content of 27.98%. The gene order was identical to the gene orders in *S. plumosum*. The putative secondary structures of ribosomal and transfer RNAs are presented in Appendix A. The mitochondrial gene order in *O. dogieli* completely corresponds to that of *S. plumosum*.

All the initial codons for 13 PEGs of *O. dogieli* were the canonical putative start codon ATG. At the same time, *S. plumosum* demonstrated a much higher diversity of start codons: ATG for ATP6, NAD4L, NAD4, COX1, COX2, ATP8, NAD6, and COB; ATT for NAD5 and NAD3; ATA for NAD1 and NAD2; and ATC for COX3. The typical stop codon TAA was characteristic for ATP6, NAD4L, NAD4, NAD1, NAD3, COX2, ATP8, and COX3 in both species as well as for COX1 in *S. plumosum* and NAD2 in *O. dogieli*. The terminal codon TAG was found in COX2 in both species. The TAA stop codon was completed by the addition of 3′ A residues to the mRNA in NAD5, NAD6, and COB in both species as well as in NAD2 in *S. plumosum* and COX1 in *O. dogieli*. Detailed information about mitochondrial protein-encoding genes is provided in Table 2.

To study the evolutionary relationships of *S. plumosum* and *O. dogieli*, we performed a phylogenetic analysis based on 13 PEGs of 17 mitochondrial genomes of siboglinids (Figure 3). Bayesian inference (BI) and maximum-likelihood (ML) analyses showed identical trees with 1.0 Bayesian posterior probabilities and 100% ML bootstrap support values for all nodes in the frenulata clade. The phylogenetic reconstruction showed the sister position of frenulates to the rest of the group uniting *Osedax*, *Sclerolinum*, and vestimentiferans. The obtained results indicated the close relationships of *S. plumosum* and *S. ekmani*, while *O. dogieli* was a sister to *S. fiordicum*.

The single-gene phylogenetic analyses of frenulates based on 16S rRNA and COX1 datasets resulted in the same topologies (Figure 4). Both analyses recovered *O. dogieli* as a sister to *S. fiordicum* and *S. plumosum* as a sister to *S. ekmani*. The ML analysis of the COX1 dataset indicated that all previously published sequences of *Oligobrachia* formed a monophyletic clade which is, together with *B. gadensis*, a sister to other frenulates. At the same time, different *Siboglinum* species did not form a monophyletic clade.

## 4. Discussion

### 4.1. Structure of the Assembled Mitochondrial Genomes

The siboglinid mitochondrial genome generally exhibits a conservative gene order, which contrasts with some annelid and mollusca representatives [15,29]. Here, we reported two new mitochondrial genomes of *S. plumosum* and *O. dogieli* belonging to the family Siboglinidae. The general structure of *S. plumosum* and *O. dogieli* mitochondrial genomes, as with most Siboglinidae and even Metazoa, are unicircular DNA molecules of about 14–16 kbp that encode the same set of genes. The mitochondrial gene order in these species is completely consistent with that characteristic of other siboglinids. The conservative gene order suggests a slower rate of rearrangements compared to other annelids. 

According to our phylogenetic analysis, a closer relationship between *S. fiordicum* and *O. dogieli* was revealed, and their genomes also possessed some similarities. It is important to note that *S. fiordicum* has an essentially larger mitochondrial genome (19 kbp) compared to other siboglinids [15], while the *O. dogieli* incomplete mitogenome has a size of 16 kbp. The total length of the *S. fiordicum* control region is more than 4 kbp. It is substantially larger than that of any of the other siboglinids. The mitochondrial genome of *O. dogieli* is partial, with an incomplete control region. Such incomplete mitogenomes are common when sequencing siboglinids [15,30], often due to the large number of small repeats and technical features of some sequencing methods. Thus, we can suggest that the overall size of the *O. dogieli* control region may be about 1.5 kbp, or even larger. All mitochondrial PEGs of the frenulates studied to date uniformly have ATG as the start codon. This is the case for *O. dogieli*, but not for *S. plumosum*, which has ATG as a start codon in most genes and alternatives such as ATA, ATT, and ATC in some genes. However, among metazoan genomes, the more varied start codon combinations are very common [15,30]. Additionally, incomplete stop codons, represented by either a single T or TA, are characteristic of some PEGs within the examined siboglinid mitochondrial genomes. Both *S. plumosum* and *O. dogieli* have four incomplete stop codons of 13 protein-encoding genes, but the genes are not completely the same.

### 4.2. The Phylogenetic Relationships of Siboglinidae

The initial molecular phylogenetic reconstructions of the family Siboglinidae were based on the single-gene sequence analyses, specifically, nuclear 18S rDNA and mitochondrial 16S rDNA [6,31], COX1 [32]. All subsequent analyses consistently placed *S. brattstromi* and *O. rubiplumus* as sisters to the remaining well-supported monophyletic vestimentiferans [15,33]. It is noteworthy that the fundamental phylogenetic trend within the Siboglinidae group remains the same in both single-gene and mitochondrial genome analyses [15,34]. We incorporated mitochondrial genomes of *S. plumosum* and *O. dogieli* into the complete mitogenome phylogeny of siboglinids with a focus on frenulates. Our research reveals very similar topologies in siboglinids to those reported in previous analyses [6,33,34].

Our data are completely consistent with all previous mitogenome-based phylogenies showing distant relationships between *S. fiordicum* and *S. ekmani*. However, while the monophyly of siboglinids is not in doubt, the intricacies of the phylogenetic relationships within and between the genus *Siboglinum* remain subjects of ongoing discourse and investigation. To date, of the 72 species in the genus *Siboglinum*, molecular data are available for only four of them [15,32]. This makes phylogenetic reconstructions of the *Siboglinum* species incomplete. It is important to note that most previous phylogenetic analyses rely heavily on single genes due to the scarcity of available nuclear markers and complete mitogenome data. Our results, together with prior data, provide additional support to the idea that the species of the genus *Siboglinum* do not form a monophyletic clade. The single-gene analyses (16S rDNA and 18S rDNA), as well as mitochondrial genome phylogenetic studies, both show that *S. fiordicum* and *S. ekmani* are non-sister taxa [6,8,15]. All our analyses identify *S. plumosum* as a sister to *S. ekmani*. According to the morphological classification, all three species of *Siboglinum* belong to different subgenera and do not have common unique characteristics among themselves [5]. Thus, the paraphyly of the genus *Siboglinum* is confirmed by both morphological and molecular data. Establishing relationships within a genus requires molecular data from representatives of all subgenera as a minimum. The presence of a single tentacle, a defining trait for the entire genus *Siboglinum*, does not necessarily indicate its monophyly [15]. Molecular data on frenulates appear fragmentary, and the range of species is determined by their availability. This limited dataset falls short of providing a comprehensive understanding of the evolutionary and divergence rates within this group; thus, the gathering of new molecular data on other siboglinid species is required.

We obtained the first instance of the *O. dogieli* mitochondrial genome. Our phylogenetic analysis placed *O. dogieli* as the sister taxon to *S. fiordicum*. Molecular data on species within the genus *Oligobrachia* are notably scarce, with molecular phylogenetic analyses involving these species being infrequent. For instance, an analysis based on the results of the mtCOX1 gene indicated that unclassified *Oligobrachia* spp. and *O. haakonmosbiensis* were sister taxa to *B. gadensis*, while *S. fiordicum* was a sister to different species of *Spirobrachia*. In this analysis, *S. ekmani* was placed as an outgroup of most frenulates [3]. Our results partially support this general topology, excluding the position of *G. brachiosum*. Complete mitochondrial genome analysis appears to be more accurate, although it can only be performed for a limited number of species. Cross-comparing different phylogenetic analyses in frenulates poses challenges due to the inclusion of distinct sets of species and reliance on disparate molecular data.

To clarify the relationships of *O. dogieli* with other species of *Oligobrachia* we performed the phylogenetic analysis of COX1 genes currently available from the NCBI. Our results are in good agreement with a previous study by Sen et al. [3] supporting the monophyly of the *Oligobrachia* clade and revealing two discernible groups within it. Additionally, all sequences from the genus *Siboglinum* are grouped into three separate clades, confirming PEG and 16S rRNA analyses and once again indicating its paraphyletic status. The invariant position of *O. dogieli* as a sister to *S. fiordicum*, distant from other *Oligobrachia* species, underscores the need for broader taxon sampling. Our results may also suggest the need for the reevaluation of morphological characters within frenulates and highlight the lack of data on this enigmatic annelid group.

## 5. Conclusions

The assembled mitogenomes of *S. plumosum* and *O. dogieli* follow the typical organization described for the family Siboglinidae. The phylogenetic analysis of 13 PEGs showed that the sequences of *S. plumosum* and *O. dogieli* were clustered with those of other siboglinids in the frenulata clade. Our phylogenetic analyses of single genes (mt16S rRNA and mtCOX1), as well as mitochondrial genomes, places *S. plumosum* as a sister to *S. ekmani*, and *O. dogieli* as a sister taxon to *S. fiordicum*, supporting the paraphyly of the genus *Siboglinum*. Our data contribute to the knowledge of annelid mitogenomes and provide valuable information for further phylogenetic and evolutionary studies.

## Figures and Tables

**Figure 1 genes-15-00077-f001:**
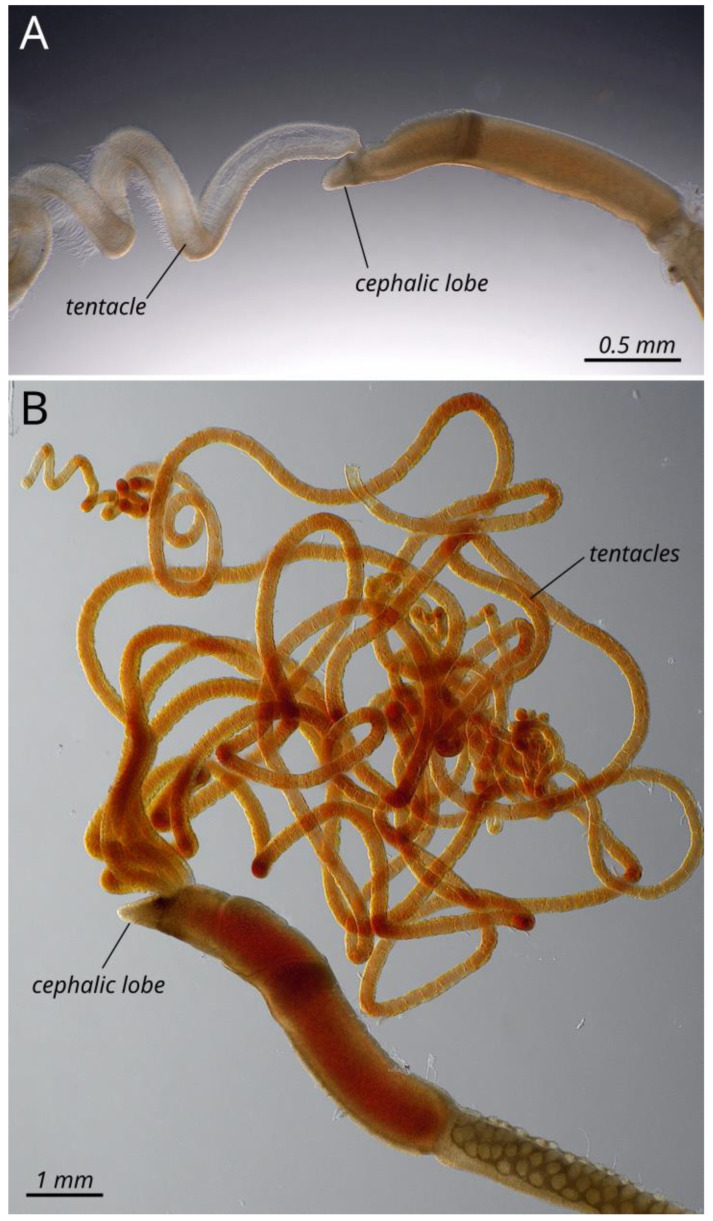
Reference images of formaldehyde-fixed adult *Siboglinum plumosum* (**A**) and *Oligobrachia dogieli* (**B**), anterior parts.

**Figure 2 genes-15-00077-f002:**
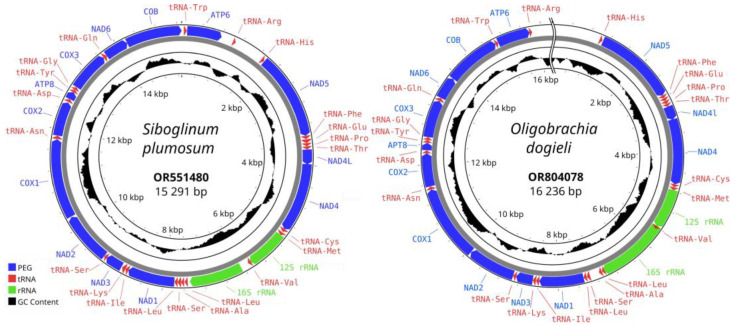
Mitochondrial genome map of *Siboglinum plumosum* and *Oligobrachia dogieli*. The 13 protein-encoding genes, 22 tRNA genes, and two rRNA genes are shown as colored blocks according to the legend provided.

**Figure 3 genes-15-00077-f003:**
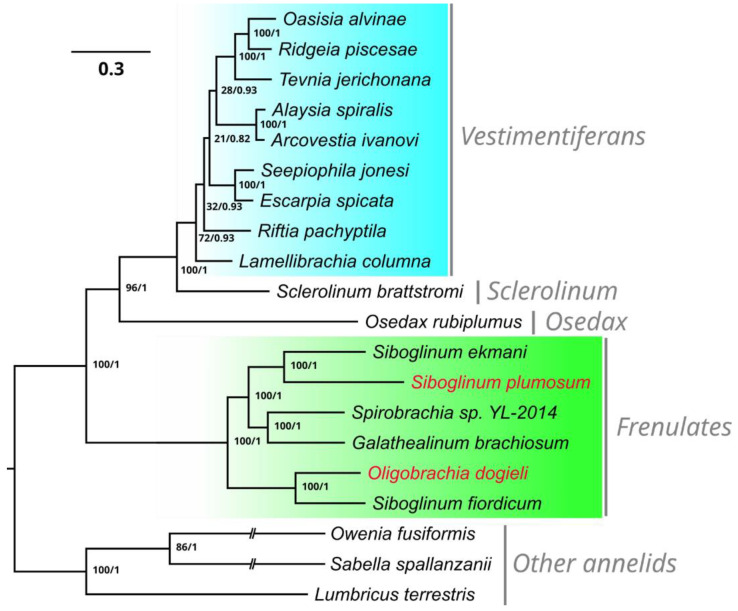
Phylogenetic reconstruction of Siboglinidae based on the 13 mitochondrial protein-encoding genes. Node labels are posterior probabilities on the left and ML bootstrap support values on the right. The vertical gray bars represent the separate clades.

**Figure 4 genes-15-00077-f004:**
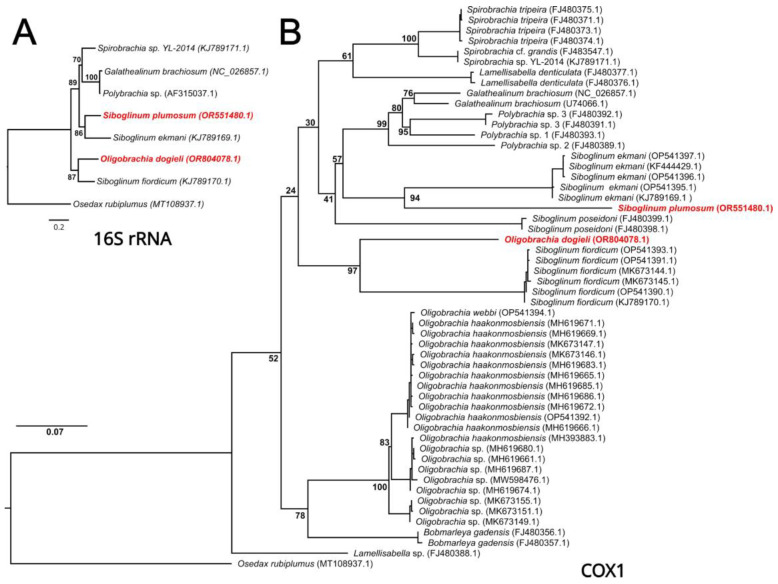
ML phylogenetic analyses of mitochondrial 16s rRNA (**A**) and COX1 (**B**) sequences from frenulate Siboglinidae. *Osedax rubiplumus* was used as an outgroup. Node labels indicate the bootstrap support values (%). The accession numbers are provided in parentheses after the sample names.

**Table 1 genes-15-00077-t001:** Partitions and corresponding models were used in the analysis. ML—Maximum likelihood; BI—Bayesian inference.

Gene	Model (ML)	Model (BI)
ATP6	GTR + F + G4	nst = 6 rates = gammangammacat = 4
ATP8	TPM2u + F + I + G4	nst = 2 rates = invgamma ngammacat = 4
COX1	GTR + F + R4	nst = 6 rates = gamma ngammacat = 4
COX2	GTR + F + R4	nst = 6 rates = gamma ngammacat = 4
COX3	GTR + F + I + G4	nst = 6 rates = invgamma ngammacat = 4
CYTB	TIM2 + F + I + G4	nst = 6 rates = invgamma ngammacat = 4
NAD1	TIM2 + F + I + G4	nst = 6 rates = invgamma ngammacat = 4
NAD2	TIM2 + F + I + G4	nst = 6 rates = invgamma ngammacat = 4
NAD3	TIM + F + I + G4	nst = 6 rates = invgamma ngammacat = 4
NAD4	TIM + F + I + G4	nst = 6 rates = invgamma ngammacat = 4
NAD4L	TIM2 + F + G4	nst = 6 rates = gamma ngammacat = 4
NAD5	TIM2 + F + I + G4	nst = 6 rates = invgamma ngammacat = 4
NAD6	HKY + F + I + G4	nst = 2 rates = invgamma ngammacat = 4

**Table 2 genes-15-00077-t002:** List of the protein-encoding genes in the mitochondrial genomes of *Siboglinum plumosum* and *Oligobrachia dogieli*. Start—the first position along α strand; Stop—the last position along α strand; Length—the size of the sequence; fcd—start codon; scd—stop codon. Asterisks label the cases where TAA stop codon is completed by the addition of 3’ A residues to the mRNA.

Gene	Length	fcd	scd	Length	fcd	scd
	*Siboglinum plumosum*			*Oligobrachia dogieli*		
*ATP6*	681	ATG	TAA	681	ATG	TAA
*NAD5*	1681	ATT	TAA *	1690	ATG	TAA *
*NAD4L*	288	ATG	TAA	288	ATG	TAA
*NAD4*	1350	ATG	TAA	1353	ATG	TAA
*NAD1*	921	ATA	TAA	924	ATG	TAA
*NAD3*	339	ATT	TAA	354	ATG	TAA
*NAD2*	990	ATA	TAA *	996	ATG	TAA
*COX1*	1548	ATG	TAA	1544	ATG	TAA *
*COX2*	675	ATG	TAG	687	ATG	TAG
*ATP8*	153	ATG	TAA	156	ATG	TAA
*COX3*	777	ATC	TAA	780	ATG	TAA
*NAD6*	466	ATG	TAA *	469	ATG	TAA *
*COB*	1092	ATG	TAA *	1135	ATG	TAA *

## Data Availability

The data that support the findings of this study are available in GenBank (https://www.ncbi.nlm.nih.gov/ accessed on 16 November 2023) under accession no. OR551480 (*Siboglinum plumosum*) and OR804078 (*Oligobrachia dogieli*). The corresponding BioProjects are PRJNA1023464 and PRJNA1039743, respectively. The raw data supporting the conclusions of this article will be made available by the authors on request.

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
