# Peer review of "The Mitochondrial Genomes of Siboglinum plumosum and Oligobrachia dogieli (Annelida: Siboglinidae) and Their Phylogenetic Analysis"

_genes, 2024, doi:10.3390/genes15010077_

Round 1
Reviewer 1 Report
Comments and Suggestions for Authors
The authors of the manuscript, “The mitochondrial genomes of Siboglinum plumosum and Oligobrachia dogieli (Annelida: Siboglinidae) and their phylogenetic analysis”, have done a great job presenting the novel mitochondrial genomes and providing phylogenetic context. I have a few comments and minor editing suggestions.
My first suggestion is to clarify the use of Frenulata and other clade names in the manuscript (Figure 3). Frenulata and Vestimentifera are both unaccepted on WORMS, and classified as equivalent to Siboglinidae, which is confusing, since the authors are frequently referring to Frenulata as a subclassification. I am not experienced in polychaete taxonomy, so I do not know if it is still appropriate to use these names; I usually don’t use names unaccepted on WORMS. However, since the authors wish to use a clade classifier between family and genus, I suggest that they use these names in lower case and make the distinction clear in the beginning. This also becomes a problem when considering the number of reference sequences, line 85 vs line 126, where only 4 mitogenomes exist yet 15 are used in the phylogeny. Doesn’t appear consistent, but the authors are discussing different levels of classification.
I am concerned with the preservation method and order of how the specimen were taxonomically identified. Previous literature (https://doi.org/10.1186/s10152-017-0501-3; doi:10.1017/S002531540705223X) has discussed the loss of features of polychaetes on preservation with ethanol, which could impact the taxonomic identification. Since it appears the authors preserved the specimen and identified them taxonomically at a later date, this needs to be addressed. Has work been done on these species before looking at how ethanol would impact identification? What features might be affected and are they important in this clade? What key/publication/features were used by Roman Smirnov (please cite)? When were the presented photographs taken, before or after preservation? This is an important issue to address because the primary goal here is to develop a molecular reference. If the taxonomy is in question, this would hamper efforts severely for confidence.
As mentioned in the manuscript, there are other single gene reference sequences in NCBI, including for a different species (S. poseidoni [COI], and several unclassified species [COI, 18S, 16S]). I suggest that individual gene trees including the two new species are produced to compare to what is currently available (supplemental figures). This would require assembling the WGS data to try to pull the 18S-28S-ITS region, but that would also be beneficial to using these new species for reference. It would be good to compare the topology of all these individual gene trees and the main mitogenome figure.
Minor comments:
· Lines 12-13: In abstract, spell out the genus names.
· Line 20: “organization” is an awkward word choice. Do they mean “adaptation” or “phenotype”?
· Line 22: add “Sedentaria subclass”
· Lines 32-37: Each of these sentences needs a reference citation.
· Line 35: What kind of chemoautotrophic bacteria?
· Line 41: “scarcity”
· Line 45: Italicize genera (here and elsewhere) [e.g. Line 48,]
· Line 48: First use of species should be spelled out with authority listed (don’t wait for line 89).
· Line 52: More accurately, “16S mitochondrial ribosomal RNA (rRNA) and 18S nuclear rRNA genes”
· Line 53: First use for these species, need to spell out genera.
· Line 56: An awkward sentence. Please revise for meaning.
· Line 59: “The tentacle number one of the morphological traits…” is not a complete sentence.
· Line 61: First use for the species, need to include the authority.
· Line 64: Unsure whether they mean “has few” or “has a few”, which would have different meanings.
· Lines 89-90: Authority should be on first use.
· Line 90: “We present…phylogenetic assessment…”?
· Line 91: “about the”
· Line 110: I believe they are talking about cleaning up the short-cycle indexed gDNA fragments, and not an amplicon product. I would just say that the “resulting indexed libraries were…”.
· Line 138 and elsewhere: Usually the acronym PEG is used for protein encoding genes.
· Line 150: Add “data not shown” to end of sentence.
· Line 168: rRNA has already been defined.
· Line 175: Italicize species names.
· Figure 2: PCG here should be changed to PEG
· Line 178: Please provide average coverage for each mitogenome.
· Table 2: If the authors are going to compare start and stop positions, I suggest that the complete genome should start numbering in the “control” region so that the start of the tRNA-His in both mitogenomes starts in the same place.
· Line 220: Add a comma between “analysis” and “closer”
· Line 223: A bit of a mismatched comparison, since the O. dogieli mitogenome is incomplete. I would at least mention it here.
· Line 237: “of”
· Line 247: “our data completely consistent” is not grammatically correct.
· Line 273: Hanging “Oligobrachia sp.”, not sure the reference. If to many unclassified species, should say that. I.E. “unclassified Oligobrachia spp.”
· Line 274: Italicize the Spirobrachia genus.
· Line 286: I’m not sure the authors feel comfortable doing this, but doesn’t this analysis suggest that the S. fiordicum should be recharacterized?
Comments on the Quality of English LanguageMinor editing suggestions as provided in the list above.
Author Response
We are grateful to the reviewer for the valuable comments and suggestions that helped us to improve the manuscript. Below you will find a point-by point answers to the comments.
My first suggestion is to clarify the use of Frenulata and other clade names in the manuscript (Figure 3). Frenulata and Vestimentifera are both unaccepted on WORMS, and classified as equivalent to Siboglinidae, which is confusing, since the authors are frequently referring to Frenulata as a subclassification. I am not experienced in polychaete taxonomy, so I do not know if it is still appropriate to use these names; I usually don’t use names unaccepted on WORMS. However, since the authors wish to use a clade classifier between family and genus, I suggest that they use these names in lower case and make the distinction clear in the beginning. This also becomes a problem when considering the number of reference sequences, line 85 vs line 126, where only 4 mitogenomes exist yet 15 are used in the phylogeny. Doesn’t appear consistent, but the authors are discussing different levels of classification.
We appreciate the reviewer for noting this. Indeed, “frenulata” has now unaccepted status. Nevertheless, it is somehow traditional and many specialists use this name for the corresponding clade. We agree with the reviewer and we have changed "frenulata" and "vestimentifera" to lower case.
I am concerned with the preservation method and order of how the specimen were taxonomically identified. Previous literature (https://doi.org/10.1186/s10152-017-0501-3; doi:10.1017/S002531540705223X) has discussed the loss of features of polychaetes on preservation with ethanol, which could impact the taxonomic identification. Since it appears the authors preserved the specimen and identified them taxonomically at a later date, this needs to be addressed. Has work been done on these species before looking at how ethanol would impact identification? What features might be affected and are they important in this clade? What key/publication/features were used by Roman Smirnov (please cite)? When were the presented photographs taken, before or after preservation? This is an important issue to address because the primary goal here is to develop a molecular reference. If the taxonomy is in question, this would hamper efforts severely for confidence.
During the sample collection a number of animals of each species were obtained. Thus we were able to use different fixation methods. We had no problems in identifying the animals and although the morphological preservation in ethanol was slightly worse than in 4% paraformaldehyde solution, all the characters important for correct identification were in good condition. To identify the specimens, the original descriptions by Ivanov (1957, and 1963) as well as revisions of genera Siboglinum (Smirnov, 2014, https://doi.org/10.31610/trudyzin/2014.318.1.48) and Oligobrachia (Smirnov, 2014, https://doi.org/10.1080/17451000.2013.872799) were used. The reference photos were made from identical formaldehyde-fixed animals from the same sample.
As mentioned in the manuscript, there are other single gene reference sequences in NCBI, including for a different species (S. poseidoni [COI], and several unclassified species [COI, 18S, 16S]). I suggest that individual gene trees including the two new species are produced to compare to what is currently available (supplemental figures). This would require assembling the WGS data to try to pull the 18S-28S-ITS region, but that would also be beneficial to using these new species for reference. It would be good to compare the topology of all these individual gene trees and the main mitogenome figure.
We appreciate the reviewer's suggestion to build individual gene trees with more siboglinid species included in the analyses. In the revised version of the manuscript, we performed two single-gene phylogenetic analyses based on COI and 16S markers. In single-gene analyses, we were focused only on the genera closely related to S. plumosum and O. dogieli to ensure better resolution of their placement within siboglinids. Species/isolates used in the analyses are listed in the Supplementary Table S1. Osedax rubiplumus was chosen as an outgroup. The resulting tree topologies match the topology of the PEGs tree placing S. plumosum as a sister taxon to S. ekmani and O. dogieli as a sister taxon to S. fiordicum. New figures with COI and 16S trees were included in the Supplementary Materials. We also revised the Methods section by adding the information about these analyses: “Separate single-gene phylogenetic analyses were performed using the newly sequenced COI and 16S data for S. plumosum and O. dogieli and data for other members of Lamellisabella, Spirobrachia, Siboglinum, Galathealinum, Polybrachia, Oligobrachia and Bobmarleya available from GenBank (Supplementary Table S1). O. rubiplumus was chosen as an outgroup. Sequences were aligned with mafft (L-INS-i method) and the resulting alignments were used to construct the ML trees in IQ-TREE with 10,000 ultrafast bootstrap replicates.”
We have carefully considered the suggestion to build an 18S gene tree using the newly obtained data from S. plumosum and O. dogieli, however, we have decided not to include this analysis in the current manuscript. While the COI marker was sequenced for several members of Lamellisabella, Spirobrachia, Siboglinum, Galathealinum, Polybrachia, Oligobrachia, and Bobmarleya, leading to a significantly improved resolution of this part of the siboglinid tree, the 18S marker is only available for Oligobrachia sp., Polybrachia sp., and Spirobrachia sp., as well as for species with fully sequenced mitogenomes. Consequently, the resolution of the 18S tree would still be incomplete. Given that the mitochondrial PEGs tree, COI tree, and 16S tree demonstrated similar topology with sufficient support, we anticipate the 18S tree to exhibit a similar pattern. Acknowledging the challenges associated with the fact that the sequencing was not initially designed for 18S analysis (e.g., the use of special primers, 18S amplification, etc.), in the revised version of the manuscript we have opted to construct only two additional gene trees for COI and 16S markers.
Minor comments:
- Lines 12-13: In abstract, spell out the genus names.
A: done
- Line 20: “organization” is an awkward word choice. Do they mean “adaptation” or “phenotype”?
A: Yes, that is right, changed to “adaptations”
- Line 22: add “Sedentaria subclass”
A: Corrected
- Lines 32-37: Each of these sentences needs a reference citation.
A: The needed citations were added to the revised manuscript
- Line 35: What kind of chemoautotrophic bacteria?
A: These are sulfur-oxidizing bacteria typical for Siboglinidae. The following reference was added.
- Line 41: “scarcity”
A: Corrected
- Line 45: Italicize genera (here and elsewhere) [e.g. Line 48,]
A: Thank you for noting, corrected
- Line 48: First use of species should be spelled out with authority listed (don’t wait for line 89).
A: Corrected
- Line 52: More accurately, “16S mitochondrial ribosomal RNA (rRNA) and 18S nuclear rRNA genes”
A: Corrected
- Line 53: First use for these species, need to spell out genera.
A: Corrected
- Line 56: An awkward sentence. Please revise for meaning.
A: The sentence was removed
- Line 59: “The tentacle number one of the morphological traits…” is not a complete sentence.
A: The sentence was removed during the revision.
- Line 61: First use for the species, need to include the authority.
A: Corrected
- Line 64: Unsure whether they mean “has few” or “has a few”, which would have different meanings.
A: Corrected
- Lines 89-90: Authority should be on first use.
A: Corrected
- Line 90: “We present…phylogenetic assessment…”?
A: Corrected
- Line 91: “about the”
A: Corrected
- Line 110: I believe they are talking about cleaning up the short-cycle indexed gDNA fragments, and not an amplicon product. I would just say that the “resulting indexed libraries were…”.
A: Corrected.
- Line 138 and elsewhere: Usually the acronym PEG is used for protein encoding genes.
A: Corrected
- Line 150: Add “data not shown” to end of sentence.
A: Corrected
- Line 168: rRNA has already been defined.
A: Corrected
- Line 175: Italicize species names.
A: Corrected
- Figure 2: PCG here should be changed to PEG
A: Corrected
- Line 178: Please provide average coverage for each mitogenome.
A: The requested average coverage data were provided in the revised manuscript. It was 709.6 for S. plumosum and 997.5 for O. dogieli.
- Table 2: If the authors are going to compare start and stop positions, I suggest that the complete genome should start numbering in the “control” region so that the start of the tRNA-His in both mitogenomes starts in the same place.
A: We decided to remove these columns from the table 2.
- Line 220: Add a comma between “analysis” and “closer”
A: Corrected
- Line 223: A bit of a mismatched comparison, since the O. dogieli mitogenome is incomplete. I would at least mention it here.
A: Corrected
- Line 237: “of”
A: Corrected
- Line 247: “our data completely consistent” is not grammatically correct.
A: Corrected
- Line 273: Hanging “Oligobrachia sp.”, not sure the reference. If to many unclassified species, should say that. I.E. “unclassified Oligobrachia spp.”
A: Corrected
- Line 274: Italicize the Spirobrachia genus.
A: Corrected
- Line 286: I’m not sure the authors feel comfortable doing this, but doesn’t this analysis suggest that the S. fiordicum should be recharacterized?
A: Currently we do not want to make such far-going assumptions. Certainly, the genus Siboglinum needs to be revised and some species should be recharacterized. However, in our opinion such revision requires stronger morphological and molecular analyses including sequencing of several representatives from different localities and broader species sampling as well. Here we only want to outline the general problem.
Reviewer 2 Report
Comments and Suggestions for Authors
This manuscript depicted the mitogenomic structures and analyzed the phylogenetic relationships of two annelids collected in the Sea of Okhotsk. I think the work is an important supplement to the genetic data of Frenulata species. But there is little description on mitochondrial structure throughout the text, excepting for the protein-coding genes. This is far from sufficient. Besides, some font and spelling errors are also detected. For examples, line 86,“Siboglinum” should be in italics; line 91, the first letter of “frenulate” should be capitalized; lines 175 and 230, the scientific names should be in italics.
Comments on the Quality of English LanguageThe author needs to carefully revise the English expression and pay attention to details.
Author Response
We thank the reviewer for the comments. We have substantially revised the manuscript and provided some additional information (coverage information, tRNA secondary structures and single gene phylogenetic analyses). We also made English proofreading and corrected a number of mistakes and typos. The spelling errors were also corrected. However, we decided to follow the comments of the reviewer 1 concerning the naming of “frenulata” and “vestimentifera”. Since these they both are unaccepted but widely used, we decided to provide them here in lower case.
Round 2
Reviewer 2 Report
Comments and Suggestions for Authors
The authors have revised the paper according to the reviewer's suggestions.
Comments on the Quality of English LanguageEnglish writing has been improved.